# Learning low-dimensional generalizable natural features from retina using a U-net

Siwei Wang[1], Benjamin Hoshal[1], Elizabeth A de Laittre[2], Olivier Marre[3], Michael J Berry II[4], and Stephanie E Palmer[1]

[1]Department of Organismal Biology and Anatomy, University of Chicago
[2]Committee on Computational Neuroscience, University of Chicago
[3]Sorbonne Université, INSERM, CNRS, Institut de la Vision
[4]Princeton Neuroscience Institute, Princeton University

## Abstract

Much of sensory neuroscience focuses on presenting stimuli that are chosen by the experimenter because they are parametric and easy to sample and are thought to be behaviorally relevant to the organism. However, it is not generally known what these relevant features are in complex, natural scenes. This work focuses on using the retinal encoding of natural movies to determine the presumably behaviorally-relevant features that the brain represents. It is prohibitive to parameterize a natural movie and its respective retinal encoding fully. We use time within a natural movie as a proxy for the whole suite of features evolving across the scene. We then use a task-agnostic deep architecture, an encoder-decoder, to model the retinal encoding process and characterize its representation of "time in the natural scene" in a compressed latent space. In our end-to-end training, an encoder learns a compressed latent representation from a large population of salamander retinal ganglion cells responding to natural movies, while a decoder samples from this compressed latent space to generate the appropriate future movie frame. By comparing latent representations of retinal activity from three movies, we find that the retina has a generalizable encoding for time in the natural scene: the precise, low-dimensional representation of time learned from one movie can be used to represent time in a different movie, with up to 17 ms resolution. We then show that static textures and velocity features of a natural movie are synergistic. The retina simultaneously encodes both to establishes a generalizable, low-dimensional representation of time in the natural scene.

## 1 Introduction

The flexibility and computational power of convolutional neural networks (CNNs) has helped sensory neuroscience model the neural code for natural stimuli with rich feature repertoires. It has been shown that CNNs carryout encoding computations similar to those observed in the retina [1, 2]. A CNN also makes the inverse problem of decoding complex stimuli from the retinal response [3] more tractable. Understanding decoding has its own significant merit because neural systems downstream of the retina can only 'see' the world through the retinal code. For any visual information to be used to guide behavior, it must first be decoded from retinal responses. Historically, the decoding problem for natural stimuli has been challenging because both the retinal response and the stimuli are high-dimensional. Although deep neural networks can capture the high dimensionality of neural inputs and responses [4], they do so by projecting the neural code into another high-dimensional parameter space that is also hard to interpret. While these tools tell us that we *can* decode, our ability to understand the features of neural activity relevant for that decoding is limited. In this work, we

36th Conference on Neural Information Processing Systems (NeurIPS 2022).

propose a novel artificial neural architecture that can decode complex natural scenes from retinal responses with high fidelity while providing a low-dimensional latent space that is interpretable. Using this architecture, we obtain unique insights into what features are important for reading out the retina's population code, and why encoding these features might enable an animal to navigate a complex, dynamic natural environment.

We anchor our results on a unique dataset from the salamander retina. Nearly one hundred output ganglion cells were recorded simultaneously while several different natural movies were projected onto the photoreceptor layer. The relatively long (20 s) movie clips were played many times, in a pseudo-random order. During the lifespan of a salamander, it goes through a transition from being aquatic to terrestrial. The sampled movies attempt to span these different motion environments. A movie of small fish in an aquarium with live plants was set up to match what a larval tiger salamander might see underwater while it hunts for food. A movie of leaves blowing in the wind resembles the scene a salamander may live in after it undergoes metamorphosis. Does a salamander retina re-use how it encodes features during the aquatic larval phase to represent features in a terrestrial scene? This motivates us to investigate whether the encoding of natural features from one particular movie is generalizable to a novel movie.

Natural movies contain complex spatio-temporal features on multiple scales. This makes enumerating all possible stimulus states in natural movies intractable. It is more feasible to investigate how the retina encodes time in the natural scene, as has been done in other studies [5]. The salamander retina elicits precisely timed spikes [6]. The idea that the retina may encode how features change over time to discriminate between frames has been explored before [7]. It has an intricate connection to stimulus-dependent representational drift in sensory systems [8]. Previous work [5] reported that a low dimensional compressed representation of activity from a mouse V1 population can be used to discriminate frames that are 1s apart. Furthermore, the authors showed that if such an encoding of time in the natural scenes exists, it is likely to be low-dimensional [5]. We train our deep neural network (an encoder-decoder in machine learning parlance) (Fig 1 and Section 2) for decoding. It reconstructs movie frames from retinal responses. This is different from previous works [1, 2] that use CNNs as "in-silico retinas" to understand how specific stimuli drive the retina. During training, the encoder part of the deep neural network learns a continuous, compressed latent representation of the retinal responses from which the decoder part samples to reconstruct target movie frames. We enforce structural constraints [9, 10, 11, 12, 13] to obtain meaningful, continuous latent space. We find that using this compressed representation, we can decode time in the natural scene up to single-frame resolution (17ms) (Fig 1). This learned, compressed representation allows for precise decoding in a novel set of natural scenes. We show that the retina is responding to spatio-temporal features that change over time, rather than having a clock. We divided these features into static (texture) and dynamic (optic flow–a vector field describing the motion between subsequent frames [14]) motifs. This allows us to construct two distributions which reflect how static features and dynamic features cluster to discriminate between frames. We find that these features are synergistic with respect to the encoding. By simultaneously encoding static and dynamic features, the retina establishes a generalizable, low-dimensional representation of time in natural scenes.

## 2  Data and encoder-decoder architecture

**Data**: Our dataset contains retinal recordings of 93 cells responding to repeated, 20 second presentations of three natural movies at 60 frames per second. There are 85-90 presentations of each movie interleaved in pseudo-random order. Spikes are binned at 17ms, to align with the movie frame duration. We compute each neuron's firing rate as a function of time within each movie, (the peri-stimulus time histogram, or PSTH) by averaging spikes across trials in these bins [1].

**Encoder-decoder architecture**: We use a U-net [15] as the backbone architecture. The U-net supplements an encoder of contracting layers with a nearly symmetric decoder of expansive layers (hence the U-shape, see Supplementary Information) [16, 17]. It is successful in domain-conversion problems, i.e., text-to-speech [18]. We modify its skip connections to supply the decoder noisy features from the encoder. Because the intermediate features from the feedforward encoder have different resolutions at their convolutional layers, these skip connections enable the decoder to form a multi-scale and multi-level feature representation of the input. Our particular network has an

---

[1]You can find our Pytorch implementation at `https://github.com/sepalmer/VU-net`

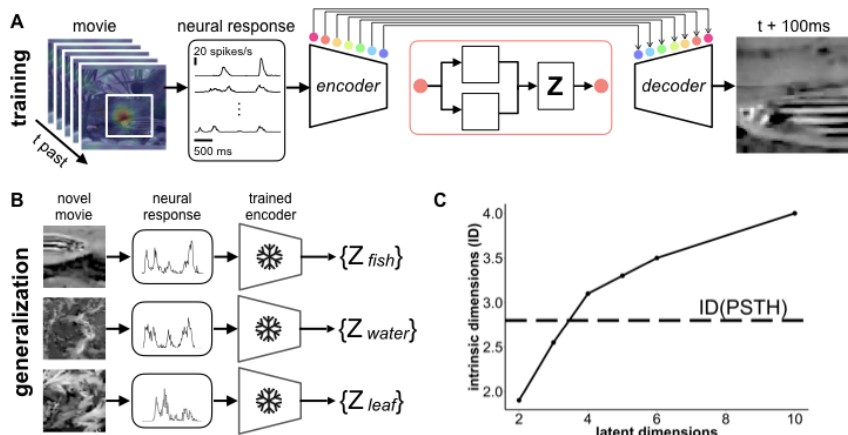

**Figure 1: A) Encoder-decoder network trained to predict a movie frame 100ms in the future from a 500ms window of retinal activity in the past (the aggregated receptive field of the retina population is shown in the heat map from the left). The network learns a low dimensional variational representation, where $\mu$ and $\sigma$ characterize the posterior distribution $p(x|z)$. The samples drawn from this latent space are referred to as $Z$ in subsequent sections. All skip connections are trained to obtain a separate latent space for $Z$. Using a highly expressive encoder [22], i.e., ResNet18, we empirically observe that the latent space learned for all skip connections are similar. B) We train the network on one movie (fish, for example), then the encoder weights are frozen. We obtain the $Z$'s for retinal activity responding to different movies by passing test samples from all three movies through the trained encoder. C) The intrinsic dimensionality of the retinal activity (dashed line) and the latent dimensions for latent space with varying $\dim Z$. Note that intrinsic dimension measures the complexity of the retinal activity. We use it as a lower bound to constrain $\dim Z$ and add additional latent dimensions to improve reconstruction of target movie frames. We stop at $\dim Z = 10$ because we observe highly accurate reconstruction with $\dim Z = 10$ empirically.**

encoder with the same architecture as ResNet18. We also initialize it with the ResNet18 [19] weights pre-trained by ImageNet [20]. Our decoder mirrors the feedforward architecture of the encoder (see Supplementary Information). In addition, we turn these skip connections into variational sampling layers and use them as compressed representations of the input retinal activity. The set of activation values in this latent space in response to the neural activity (input) from a given stimulus movie is referred to as $Z_{\text{movie}}$, where 'movie' is either 'fish', 'water', or 'leaf'). We constrain all variational sampling layers to have the same dimensionality, to simplify the training.

Knowing that the decoder reconstructs the movie frame from features across multiple spatial and temporal scales, we use the perceptual loss [21] as the objective function for the encoder-decoder. This loss function compares a reconstructed frame with its target frame with respect to learned features from a pretrained VGG, as opposed to their raw pixels. Throughout the paper, we use features from the pretrained VGG19 to represent movie frames.

## 3 Results

### 3.1 Retinal activity has a low intrinsic dimension

Fig 1A shows how we train an encoder-decoder for a specific movie (in this example the fish movie). The input is the retinal PSTHs from the 45 most reliably spiking cells in the population (see Supplementary Information). We take trial-averaged firing rates, and ignore 'noise' correlations between cells, to make an initial pass at this high-dimensional problem. Considering that many of the observed retinal computations happen on a timescale below 400ms [23], we restrict ourselves to 500 ms long snippets of the PSTHs. To encourage the model to learn temporal structure within the movie, we ask the decoder to reconstruct a movie frame 100ms in the future, after the end of the 500ms snippet of neural response. We choose this particular $\Delta t$ based on the timescale of predictive information in retinal populations [24]. We train one predictive encoder-decoder for a specific movie using 40,000 training samples (see Supplementary Information for details) with a 90%/10% training/validation split. The reconstruction is from an additional held-out test set of 10,000

samples (100 frames, 100 PSTH patterns in each frame, see Supplementary Information). We also train a static encoder-decoder that learns to reconstruct a frame centered within the 500ms window of neural activity (using 250ms before and after the target frame as the input, similar to [3]). The predictive encoder-decoder achieves a reconstruction performance similar to the static one. We focus on the predictive encoder-decoder in our subsequent analysis because it may capture both static and temporal structures of a natural movie by design.

The latent space is a compressed representation, $Z$, of the mean firing rate patterns from the retinal population. We estimate the intrinsic dimension [25, 26] of the retinal activity and use it to guide the selection of the dimensionality of the latent space. This intrinsic dimension is the number of variables needed to describe a data distribution [27]. It is also a complexity measure of data because it determines the number of samples needed to characterize a data manifold [28, 29]. We find that our estimate of the intrinsic dimensionality for all retinal activity in response to the three movies yields the same result as the estimate of the intrinsic dimensionality from the activity for each movie separately, i.e., including retinal activity from a different movie does not add complexity to the latent space representation of the retinal response. This suggests that features in natural movies may be encoded by the retina in a generalizable way across movies. We use the intrinsic dimension of the retinal activity as a lower bound to determine latent dimension, i.e., $\dim Z$ in the encoder-decoder. We then add additional latent dimensions to help encode factors that may result from combinations of different intrinsic dimensions. We observe that the increase in the estimated intrinsic dimension of the latent space decreases after $\dim Z > 4$. Meanwhile, we empirically observe good reconstruction performance (the pixel MSE is about 0.02 averaged over 100 test frames of size 64X64 with pixel intensity $\in [0, 255]$) when we use latent dimension equal to 10 to reconstruct the held-out movie segments (see Supplementary Information). Thus, for further analyses, we use $\dim Z = 10$ unless otherwise stated. To address questions about whether features learned from retinal activity responding to one movie are generalizable to another novel movie, we generate "mismatched" $Z$'s. These "mismatched" $Z$'s are compressed representations of the retinal responses to $Z_{water}$ and $Z_{leaf}$ movies by passing those inputs through the encoder trained for the $Z_{fish}$ movie (Fig 1B). The results presented below are similar regardless of which movie (water, leaf, fish) we train on and which other two movies are used to generate mismatched $Z$'s.

## 3.2 The retina encodes complex, but interpretable spatio-temporal features in natural movies

The goal in this section is to qualitatively assess latent space stimulus features to show that the encoder-decoder contains features that are plausibly encoded by the retina. We visualize highly activated features from decoding layers of multiple spatial scales (threshold by top 1% activation intensity, see Supplementary Information for more visualization). We find both features that resemble background motion and features that resemble object motion (Fig 2) in a specific decoding layer. In particular, the object motion feature closely traces the movement of the fish in the target movie segment. The average activation of this feature along the x-axis shows that this feature may be responsible to decode both position and velocity of the fish movement in the target movie frames (top row). Because removing the latent activations silences both background and motion features, we also determined that this specific variational sampling layer generates these disentangled features. This observation agrees with previous experiments that the retina can encode features that evolve across time and space [30]. This motivates us to investigate whether we can decode the "time in the natural movie", i.e., discriminate between different frames from this specific decoding layer.

## 3.3 Low dimensional, generalizable representation of time in multiple movies

In this section, we investigate whether we can discriminate frames in natural movies and thus decode time in natural scenes. We can also ask whether the feature space of the retinal population used for one movie can be used for the other two natural movies in our three-movie dataset. If this is the case, it would suggest that there is a general representation of spatio-temporal features in retinal activity that supports this decoding of time.

Inspired by previous work that the visual cortex may contain a low-dimensional representation of complex stimulus features evolving over time [5], we decode time in the natural scene with the particular latent space that corresponds to the decoding layer shown in Fig 2. We obtain the compressed representation of retinal activity on the held-out test frames for the fish movie, i.e., $Z_{fish}$, as well as test frames for two other novel movies (e.g., leaf and water). This use of an encoder-

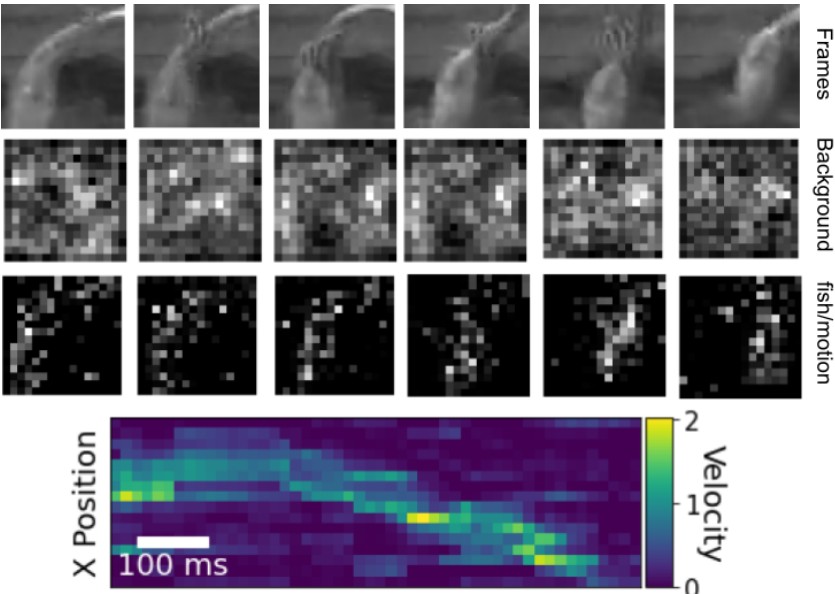

**Figure 2: Features from a decoder layer separately decode background and object motion. 1st row (from top to bottom): target movie frames that are 100ms apart; 2nd row: an example feature inside the trained decoder and its activation for decoding frames in the top row. 3rd row: the spatial, temporal activation. 4th row: X-T receptive field (aggregated x-axis activation over time).**

decoder to generate a compressed representation of a data distribution has been investigated in detail in representation learning [9]. For each $Z_{fish,leaf,water}$ separately, we linearly decode the 1D frame label of held-out test frames from the corresponding $Z$. Fig 3A shows all decoding performance as a function of the number of dimensions allowed in $Z$ (we trained a series of encoder-decoders with different $\dim Z$). Because all encoder-decoder models are trained with the fish movie, the decoding for the fish movie outperforms the other two. However, we observe that $\dim Z = 5$ is sufficient for the encoder-decoder to build a latent space $Z$ which can decode >80% frames in all three movies. This is also the dimensionality needed for decoding in the natural scene reported in mouse visual cortex [5]. Since these frames are 17ms apart, this decoding performance shows that salamander retina establishes a low dimensional representation of "time in the natural scene" with fine temporal resolution.

The high decoding performance from the mismatched $Z$'s suggests that retinal activity may establish a general encoding for "time in the natural scene". How is this possible? The retina is presumably encoding general space-time features that are predictive of the future space-time features in natural scenes. These could be complex, but they seem to generalize across movies. We use information theory to directly evaluate the information that all three $Z$s contain that is relevant for decoding time. We first calculate the mutual information between latent representations of retinal activity responding to different movies (here,$Z_{fish}$ is the learned compressed retinal responses to the fish movie, so we show the mutual information of fish vs. leaf $I(Z_{fish}; Z_{leaf})$ and fish vs. water, $I(Z_{fish}; Z_{water})$, respectively). We also observed that $I(Z_{fish}; Z_{leaf}, time) = I(Z_{fish}; Z_{leaf})$, i.e., including time does not add additional mutual information. This tells us the latent representation obtained from retinal activity for one movie encodes the generalizable "time in natural scenes" for a different movie, up to the full entropy of time itself. To show this, we use the chain rule of mutual information and subtract mutual information that is independent of time, i.e., $I(Z_{fish}; Z_{leaf}|time) = I(Z_{fish}; Z_{leaf}, time) - I(Z_{leaf}; time) = I(Z_{fish}; Z_{leaf}) - I(Z_{leaf}; time)$, in Fig 4A. The cyan bar shows the MI between $Z_{fish}$ and mismatched Z that is about time, $I(Z_{fish}; Z_{leaf}) - I(Z_{fish}; Z_{leaf}|time)$. It nearly saturates $H(time)$. We also observe that the mutual information between different movies is mostly about time. The difference between the pink and cyan bar is very small. This estimate confirms that the retina has a generalizable, precise representation of time in natural scenes that can discriminate consecutive frames that are only 17ms apart. Table 1 confirms that latent representations obtained from retinal responses to any one movie can be used to decode time for all three movies.

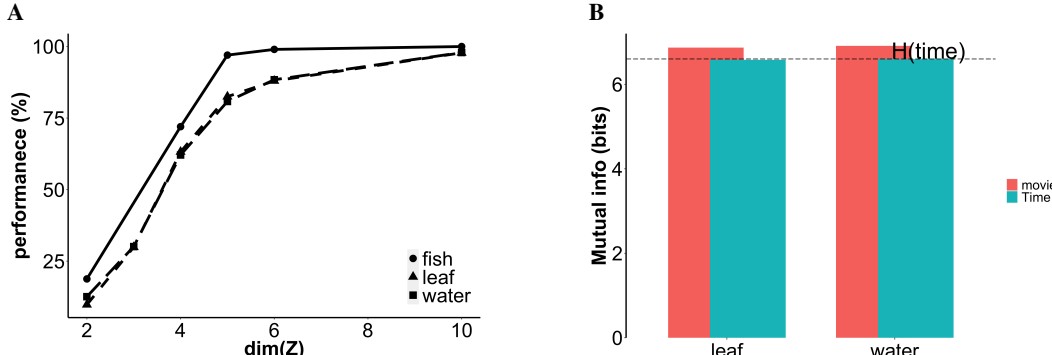

Figure 3: A) Decoding performance of latent representations on test movie segments for all three movies. The encoder-decoder is trained in fish movie. The latent representations are obtained from retinal activity of all three movies (fish, leaf, water). B) Mutual information between movies vs. mutual information with respect to time. $H(time) = 6.6$ bits for the test movie clip of 100 frames long. Because we use the encoder-decoder trained with fish movie here, we show $I(Z_{fish}; Z_{leaf})$ and $I(Z_{fish}; Z_{water})$. (See the supplementary information)

|  | Water(5d) | Water(10d) | Leaf(5d) | Leaf(10d) |
|---|---|---|---|---|
| Fish | 78.2% | 96.9% | 72.2% | 98.0% |
| Leaf | 79.5% | 97.8% | 84.4% | 99.1% |
| Water | 84.0% | 97.4% | 71.7% | 97.9% |

Table 1: Latent representations trained on any one movie can decode time in all three movies. Here we show $\dim Z = 5$ and $\dim Z = 10$, respectively.

In the supplementary information, we also included simple visualizations and decoders to demonstrate that decoding "time in the natural scene" is challenging. First, "time in the natural scene" cannot be observed as a simple visual trend using the mean PSTH, pairwise frame-to-frame distance, or the 2D latent space trace. Second, we also included linear decoders trained on instantaneous PSTH's, raw PSTH's, shuffled PSTH's and three other dimensionality reductions (Isomap, 10D-PCA and 50D-PCA). We found that shuffled PSTH shows inferior performance compared to the one trained on raw PSTH's. This suggests that decoding time does not come from trivial gross changes in spiking statistics. We also observed that linear decoders fall short of the 10-D latent representation learned by the U-net in terms of generalizable performance, even when using the 50D-PCA. To learn a low dimensional feature space that can be applied to all three natural movies, our variational U-net carries out significant nonlinear transformation.

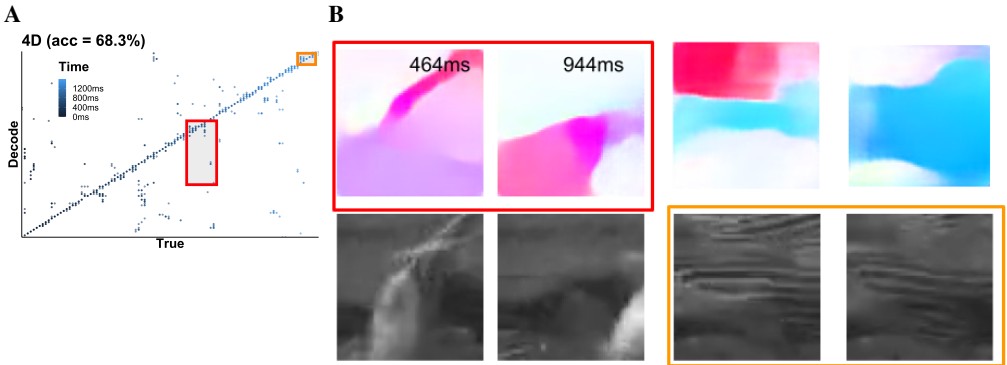

Figure 4: A) The scatter plot of decoding performance with a $Z_{fish}$ of $\dim Z = 4$. Correctly decoded samples appear along the diagonal line of true vs. pred (decode), incorrectly decoded samples show up in off-diagonal regions. With $\dim Z > 4$, the decoding performance increases to $\sim 95\%$ (shown in Fig 3). B) Two examples of decoding errors (within the boxes shown in A). The two frames in the red box contain similar optic flow (dynamic) whereas the two frames in the orange box contain similar static textures.

### 3.4 Synergistic features for encoding "time in the natural scene"

Fig 3 shows that the retina has a low-dimensional, generalizable representation for time in the natural scene. We next ask what features the retina uses for this low-dimensional, generalizable representation.

Although there are retinal circuits that encode object motion, most decoding work only uses natural images as their input stimuli [31]. Our stimulus set allows us to explore how natural texture and motion might interact. In Fig 4, we show two example decoding errors (Fig 4B), one where the predicted and true frame have similar static features and one other where the predicted and true frames with similar optic flow (dynamic features). This indicates that our observed latent representation of "time in the natural scene" can be confused when either the static or dynamic structure between frames is similar. These examples are not sufficient to exclude the possibility that static textures may also be used to discriminate between different dynamic frames. To investigate this, we perform two parallel hierarchical clusterings: one on the frames themselves (static) and one on the optic flow frames (dynamic) of the test movie segment. If textures govern the discrimination between frames in both dynamic and static settings, then these clusterings should produce similar clusters. Before clustering, we convert all static frames and their corresponding optic flow frames to features that are the activation from the last ReLu layer in a pre-traind VGG19 network [32]. These activations are believed to mimic features used in human perception of generic natural stimuli [21]. We observe that hierarchical clusterings on dynamic versus static motifs of all three movies yield different results (see Supplementary Information for details).

The difference between clusters of frames based on static versus dynamic features enables us to construct three distributions of "time in the natural scene" (Fig 5A). One uses the clustering based on the static features $Y_{static}$, another using the dynamic features $Y_{dynamic}$, and a third combining both sets of features $Y_{joint}$. By construction, we would like the joint distribution $Y_{joint}$ to have an entropy as close as possible to the full entropy of time $H(time) = \log 2(100)$=6.6 bits. This construction narrows down our search of coding schemes to how the retina *combines* dynamic and static structures within natural movies to encode the "time in the natural scene".

Previous work showed that the retina performs efficient coding [33, 34]. Efficient coding predicts that redundancy should be minimized among different features of interest. Therefore, while we would like the joint distribution $H(Y_{joint})$ to contain most of the entropy of time, we also want to minimize the mutual information between the two components $Y_{static}$ and $Y_{dynamic}$ (minimize their redundancy). We varied the threshold on the clustering hierarchies to coarse grain the distributions $Y_{static}$ and $Y_{dynamic}$, and also to create the joint distribution $Y_{joint}$. Because we do not know *a priori* their relative contributions, we coarse-grain both clusterings with thresholds such that the coarse grained entropy of the two components are comparable, i.e., $E(Y_{static}) \sim E(Y_{dynamic})$. In Fig 4A, we show the coarse-grained joint distribution we use for subsequent analysis. It contains a small amount of redundancy, i.e., $I(Y_{dynamic}, Y_{static}) = 0.8$ bits, while $Y_{joint}$ includes most of the entropy for time ($H(Y_{joint}) = 5.05$ bits = 76% of $H(time) = 6.6$ bits). These two components are dominated by either static or dynamic features, respectively. In Supplementary Information, we discuss in detail how we find these distributions. A different threshold may either introduce a significantly higher redundancy or sacrifice too much information from $H(time)$.

Using these distributions, now we can ask how the neural population encodes time through encoding the both the static and dynamic features of the natural scene. The latent activation of retinal inputs are compressed representations, so we frame this encoding problem using the information bottleneck method [35]. The information bottleneck method identifies whether a compressed representation $T$ retains as much information as possible about the relevant variable $Y$ while compressing away irrelevant components of the input $X$. In this context, the information bottleneck shows how much information a compressed representation needs, in order to encode a specific amount of information about the features of interest, $Y$.

The information bottleneck method minimizes the following objective function:

$$\mathcal{L}_{p(t|X),\beta} = I(X;T) - \beta I(Y;T) \tag{1}$$

To make a meaningful measurement of $I(X;T)$ and $I(Y;T)$, we first ensure that we have a meaningful latent representation $Z$. The challenge is to prevent the so-called "posterior collapse" [11]. This is a phenomenon previously reported in encoder-decoders with highly expressive architectures (like the

ResNet18 network that we use here) [22]. These expressive architectures are capable of decoding complex features, e.g., our movie frames, without using $Z$. This results in the latent code $Z$ only containing noise, i.e. $I(X;Z) = 0$. Here, we use a simple heuristic to circumvent this scenario. As discussed in [36, 12, 37], we can obtain a meaningful $Z$ by making the posterior to have a small but nonzero noise. To be specific, we have $\sigma^2 > 0$ for $p(X|Z)$. This is also dubbed a 'committed rate' for the encoder. In our case, we choose $\log \sigma^2 = -1.0$ to further ensure numerical stability of the mutual information estimator that we use [38].

The mutual information estimator also requires the prior of $p(Z)$ to have a factorized marginal (each marginal is an independent Gaussian), $\mathcal{N}(0, I)$ (they are independent Gaussians). This is a typical constraint introduced in the original variational autoencoder [13]. Combining this constraint on $p(Z)$ and the above constraint on $p(X|Z)$, we can approximate $I(X;Z)$ with the following estimator,

$$
\begin{aligned}
I(X;Z) &= H(Z) - H(Z|X) \\
&\leq -\frac{1}{P}\sum_i \log \frac{1}{P}\sum_j \exp(-\frac{1}{2}\frac{||z_i - z_j||_2^2}{\sigma^2}) - \frac{D}{2}(1 + \log \sigma^2 + \log 2\pi)
\end{aligned}
\tag{2}
$$

$P$ is the number of test samples ($P = 10000$ in our case) and $z_i$, $z_j$ are the latent activations for the $i_{\text{th}}$ or $j_{\text{th}}$ sample.

Because $P(Y)$ has a uniform distribution (100 retinal inputs per frame and there are 100 held-out frames), we can use the following estimator for $I(Y;Z)$ (we also validated this estimation with a widely used non-parametric estimator [39], the difference is less than 0.1 in all our calculations):

$$
\begin{aligned}
I(Y;Z) &= H(Z) - H(Z|Y) \\
&\leq -\frac{1}{P}\sum_i \log \frac{1}{P}\sum_j \exp(-\frac{1}{2}\frac{||z_i - z_j||_2^2}{\sigma^2}) \\
&\quad -\sum_{l=1}^{L} p_l \left[ -\frac{1}{P}\sum_{i,Y_i=l} \log \frac{1}{P}\sum_{j,Y_j=l} \exp(-\frac{1}{2}\frac{||z_i - z_j||_2^2}{\sigma^2}) \right]
\end{aligned}
\tag{3}
$$

Note that $p_l$ is the number of test samples for the $l_{\text{th}}$ frame. $p_l = 100$ in all of the test distributions.

Using $p(Z)$ with a factorized Gaussian marginal brings an additional benefit: it is the sufficient and necessary condition for the latent space to exhibit orthogonal symmetry [40, 41, 42]. This Darmois-Skitovitch characterization was first introduced to identify unique factors for independent component analysis [43, 44]. Recent work also showed that this factorized marginal $Z$ is necessary to capture "ground truth factors of variation" [10] in the latent representation $Z$ [45, 12].

In Fig 5B, we show the information plots for time, dynamic features, static features and the joint ($\{Y_{dynamic}, Y_{static}\}$) features. $X$ is the retinal activity. $Y$ are features of interest, which are time, $Y_{dynamic}$, $Y_{static}$ and $Y_{joint}$, shown in different colors. $Z$ are the latent activations of a series of encoder-decoders with different $\dim Z$. Because the encoding of time reaches its full entropy of $H(time)$ (specifically, we observe this saturation at $\dim Z > 5$), this shows that the compressed representations $Z$ from the encoder-decoder are near-optimal. In the information bottleneck technique, the best any representation can do is to encode the full entropy of the relevant variable $Y$, i.e., here $H(Y) = H(time)$. This is similar to the near-optimality shown in previous work in the variational information bottleneck [46, 38]. In addition, we use many $Z$'s spanning a range of dimensions to construct these information curves. We observe that all of these compressed representations encode comparable amounts of information about both dynamic and static features. This suggests that the dynamic features are as prominent as the static features in the retinal population's internal representation of complex natural scenes.

We highlight the benefit of simultaneous encoding of dynamic and static features in Fig 5C. By comparing the information about $Y_{joint} = \{Y_{static}, Y_{dynamic}\}$ and the sum of the information about these two components, we observe that there is synergy $= I(Y_{\text{joint}};Z) - (I(Y_{\text{static}};Z) + I(Y_{\text{dynamic}});Z)$. By construction, the presence of synergistic information corresponds to latent representations with lower dimensions. This suggests why the retina can compress its encoding of both static and dynamic features. When it encodes both features simultaneously, the synergy between these features helps the retina to represent the full entropy of time itself in fewer latent dimensions.

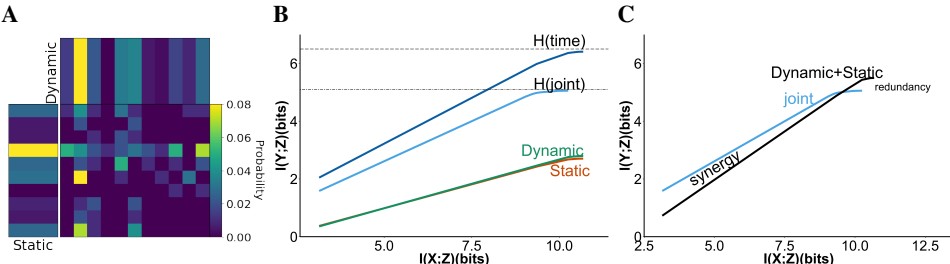

**Figure 5: A: Joint distribution of static and dynamic features.** This joint distribution includes 76% of the entropy of time. Note that the mutual information between coarse-grained static and dynamic distributions is about 0.8 bits. Given that the entropy of static/dynamic features is around 2.8-2.9 bits, the amount of mutual information between them is relatively small. **B: the information plane for fish data.** Dark blue: the information curve for encoding time; Light blue: the information curve for encoding the joint distribution combining static and dynamic features; Red/Green: information curves for separated static (red) and dynamic (green) features. See Supplementary Information for information curves of the other two movies **C:** Blue: the information curve for encoding the joint distribution, the same as B; Black: the sum of information curves from Dynamic(Red)+Static(Green). There is a synergistic region between the information curve for the joint and the sum.

# 4 Discussion

This work uses a U-net-based deep learning architecture to reverse engineer a retinal encoding process for complex natural movies. Using the PSTHs of a large salamander retinal population, we identify stereotypical features that are generalizable across multiple natural movies. We find that the retina uses a transferable, low dimensional representation to encode a rich set of natural space-time features. The encoding obtained from one movie can be used to decode "time in the natural scene" for a different movie, despite differences in their particular spatio-temporal structures. We also discover that the retina encodes time through synergistic coding of both dynamic and static features.

Here, we only observed synergy within the feature space (using mean firing rates of retinal activity, we assume all cells are independent). We also decoded time in its simplest form by asking how well we discriminate between different frames. In future work, we would like to extend our analysis to temporal structure with proper predictive constraints, i.e., predicting a future at a longer $\Delta t$ should be more challenging than predicting a smaller $\Delta t$ [47, 24]. We are also aware that the synergy here is different from what can be observed between cells in the neural data. The synergy in the neural code may combine synergy in the feature space with synergy in the population code, itself [48, 49].

Our work is most similar to [50, 51] when compared to other methods that also identify a latent representation between brain activity and external stimuli. They used a multilayer perceptron (MLP), a highly expressive feedforward encoder. MLP is fully-connected, so that its learned latent representation corresponds to a single global scale. Our U-net architecture, in contrast to the MLP, employs a ResNet as the encoder. The ResNet encoder attains the same performance as the MLP, but by cascading Resblocks from coarse-to-fine scales. This makes it possible for the U-net architecture to simultaneously learn compressed latent representation at various scales. Although we did not specifically explore this feature, it might be relevant for future research on understanding brain dynamics in flexible natural environments. For example, there is a hierarchy of timescales both in natural scenes and output natural behaviors, ranging from hundreds of milliseconds to minutes (whisking to walking to making action plans [52, 53]). With additional constraints [54], These variational sampling layers may learn hierarchically distinct latent representations for each timescale individually and comprehend how they might be coupled to create complicated behavioral outputs. Outside of neuroscience, This U-net is compatible to learn latent representations between other temporal sequences (e.g., text) and complex spatio-temporal signals (speech or video). Text-to-speech and video summarization are two possible applications. Combining latent representation at multiple scales may also reveal semantic relationships between complex features in general object recognition, e.g, how does a model combine local features (nose, eye) with global shape (e.g., body size) to discriminate between cats and dogs.

Our work shows that the retina leverages feature representations that are common across natural movies. This knowledge transfer differs from what is referred to as "transfer learning" in computer vision and machine learning. In computer vision, transfer learning refers to training a model with a much more complicated dataset (e.g. ImageNet with 1000 classes) and performing inference on a novel, but much smaller dataset (e.g. CIFAR10/100 or CelebA). Transfer learning presupposes that models trained on complex datasets contain sufficient variation to allow the learned features to be reused on new datasets. For the retina, evolutionary timescales underlie the "training from a complex dataset" stage. The retina is shaped in such a way that behaviorally significant components of all natural inputs in an organism's ecological niche are selectively encoded. This enables our training on one movie/retinal response dataset to reveal features transferable to another movie of a similar complexity or scale. Future studies may enable us to determine if such a generalizable feature representation is innate (sculpted only by evolution) or whether visual experience within a lifetime may refine it. This would depend on our ability to track changes in visual processing beyond the retina (e.g. cortex) over the course of an animal's life (similar to fine tuning in the transfer learning domain).

## Acknowledgments and Disclosure of Funding

This work was supported by the National Institutes of Health BRAIN-R01 EB026943, the National Science Foundation (through the Center for the Physics of Biological Function PHY-1734030 and Clustering of Neural Activity: A Design Principle for Population Codes PHY-1806932) and An ERC CoG grant (grant agreement 101045253) and Grants from the ANR (DECORE, ShootingStar). This work was also based upon work supported by the National Science Foundation Graduate Research Fellowship Program under Grant No. (DGE-1746045). Any opinions, findings, and conclusions or recommendations expressed in this material are those of the author(s) and do not necessarily reflect the views of the National Science Foundation.

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
