# OpenReview forum: "Learning low-dimensional generalizable natural features from retina using a U-net"
_NeurIPS.cc/2022/Conference — NeurIPS 2022 Accept_

### Official Review · Reviewer_YUEv · 2022-07-06

**Rating:** 2
**Confidence:** 4
**Soundness:** 1 poor
**Presentation:** 2 fair
**Contribution:** 1 poor

**Summary:**

In this manuscript the authors try to apply a U-net architecture to predict future movie frames from salamander retinal cell responses to the previous frames. The main finding is a signal for the time since the start of the movie in the latent representation.


**Questions:**

Unfortunately, I don't think my concerns could be removed with anything the authors could do within the confines of a rebuttal phase.

In general, find a sufficiently large dataset, find an input format for the retinal responses that has some spatial correspondence and keeps which neuron (type) responded where, and try to include some interpretability of the output to find out what can be decoded.

**Limitations:**

---

**Strengths And Weaknesses:**


Unfortunately, I don’t think any of the methods or results in this paper are useable. The plan to train a large DNN architecture on the reconstruction of a few seconds of movie is doomed from the start. A U-Net architecture can probably reconstruct the whole movie clip from a single number for the frame as input without any retinal data. This also means that finding a signal for which frame to construct is not unexpected.
Moreover, I fail to understand what the underlying aim of training this U-Net was in the first place. A deep neural network trained to predict future video frames is a technique for unsupervised learning, i.e. explicitly to learn complex features that are not easily available in the original inputs and filter out quickly changing information. I don’t see how this would be helpful for understanding the encoding of retinal cells, even if it worked.

Besides these fundamental problems, there are some odd choices in the model design and sometimes it was not entirely clear what the authors did exactly:
1) The input is a stack of “Gramian angular field” images constructed from the PSTHs for individual neurons. This transformation has no spatial correspondence to the image. Thus, the whole U-net architecture makes little to no sense with this input.
2) Similarly the weights of a resnet pretrained on images are not sensible for this kind of input.
3) The authors add noise to the architecture at various points without much justification except a hope for more interpretable representations.
4) At various places, I lost track which analysis were based on the U-Net and which ones were based on the additional VGG.
5) The training section in the supplementary material suggests that different samples were generated by dropping 2 cells randomly, but there is no explanation how the network could handle input with permuted cell identities sensibly.

---

> ### Author Response · Authors · 2022-08-02
> **Thank you for your feedback, response part I**
>
> A U-Net architecture can probably reconstruct the whole movie clip from a single number for the frame as input without any retinal data.
>
> We agree that it would be a terrible idea to suggest that a U-net needs neural data in addition to movie frames to decode natural movies. However, the focus of this paper is to understand what features are encoded by the retina alone. Using the retinal code as the only input to decode movie frames is critical to answer our neuroscience questions. U-net architecture is highly expressive. It may be overfitted to generate natural images (Ulyanov 2017), but we cannot simply feed in frames or a single number from our natural movies to decode future movie frames and expect to learn interesting aspects of the retinal encoding. In addition, all our results use a hold-out test dataset.
>
> “Find a larger dataset”: We agree with the general gist of this comment. Having a large dataset helps to quantify population coding better. We recorded this dataset ourselves because there is no publicly available dataset of the retina responding to a diverse set of natural movies. In neuroscience, using retinal data to interrogate neural code writ large is appealing because the ex vivo retina is a rare intact system during recording. In addition, local retinal spiking statistics are generalizable to much larger retinal populations. With these benefits in mind, we used recordings from a retinal population of nearly 100 cells with a collective response field  that spans 2-3 single-cell receptive field diameters (see Fig 1A and supplementary Fig 1A). To be able to compare the retinal responses from one movie vs. another, it is critical that we use data from the same retina. Inevitably, we are constrained by how long one can obtain stable responses from a single retina (~4 hours). Meanwhile, we must play each stimulus 80-100 times to obtain reliable trial-averaged measurements. This further limits the number of stimuli we can play in one recording session. Combining all these factors, this is the best dataset available to us, especially for the neuroscience questions in which we are interested.
>
> Detailed response:
>
> The input is a stack of “Gramian angular field” images constructed from the PSTHs for individual neurons. This transformation has no spatial correspondence to the image. Thus, the whole U-net architecture makes little to no sense with this input.
> Similarly the weights of a resnet pretrained on images are not sensible for this kind of input.
>
> We obtain the spatial correspondence between the retinal population and a movie frame indirectly from the receptive fields of the retinal population. We showed the aggregated receptive field in Fig 1A and their individual, densely overlapped receptive fields in supplementary Fig 1A. Throughout this paper, we focus on how the retinal population encodes the portion of natural movie within the aggregated receptive field. Throughout this study, we use raw PSTH’s from the retinal population. In neuroscience, raw PSTH’s are stereotypical neural responses to stimulus features.
>
> In our U-net architecture, we use variational sampling to replace direct skip connections. This allows us to learn a latent space with variable z and two probability distributions: i.e., an encoding distribution q(z|PSTH), and a decode distribution p(future movie frame|z). These distributions constrain the latent space (combining all variational layers) to contain factors that bridges between retinal PSTH’s and natural movies. Because PSTH’s encodes stereotypical features of natural movies, we hypothesize that such a latent space may tell us what features from natural movies are selectively encoded by the retina. Using variational sampling (the key element of the variational autoencoder) to learn a consistent latent space between different data domains or modalities is not new. Recent applications include estimating 3D joint angle coordinates from 2D RGB images of human hands (Spurr et al., 2018) and  generating audio from a corresponding (a human voice of “zero” to an  image of zero, Tian et al., 2019). Neuroscience experiments suggest that even a single cell in the retina can encode complex features (e.g., object motion, reversal, predictive features Kuhn 2016, Gollisch 2010, Schwartz 2007). This motivates us to use ResNet (a highly expressive architecture) as the backbone for our encoder. In addition, because our aim is to reconstruct frames of natural movies, we use an overall U-net encoder-decoder because its hierarchical decoder is capable of reconstructing natural images.

---

> > ### Author Response · Authors · 2022-08-04
> > **response part II**
> >
> > We agree that the Gramian angular fields do not spatially correspond to the movie frame. We construct the “Gramian angular field” images for an engineering benefit. This enables us to initialize the encoder portion of U-net with pretrained weights from ResNet18. Initializing some portion of a deep neural network model with structured weights (instead of random weights) speeds up its training. Recent work in financial forecasting (Barra et al., 2020) shows that the Gramian angular field not only captures subtle statistical structures of time series, but the engineering advantage from using a 2D image (initialization with pretrained ResNet18) is impressive. In our case, the decoder portion is always initialized with random weights. We trained the model end-to-end, so the final encoding weights are different from the pretrained ResNet18.
> >
> > Barra et al., Deep Learning and Time Series-to-Image Encoding for Financial Forecasting 2020
> >
> > The authors add noise to the architecture at various points without much justification except a hope for more interpretable representations.
> >
> > Thank you for pointing out this confusion.
> > We do not add noise to obtain more interpretable representations.
> >
> > The retinal activity is noisy by nature. Because our goal is to understand how the retina encodes stimulus features, adding noise to the architecture constrains the model to work in a more biologically plausible way.
> > From a technical perspective, adding noise is necessary to prevent posterior collapse in the variational sampling step. It also enables us to evaluate mutual information between the input (from retina) and the model’s latent representation. Otherwise, once the model is trained, it would act as a deterministic function of the input. In this case, the mutual information between the retinal activity and the latent representation is not well-defined (MI(x,f(x)) is infinite if f() is a deterministic function). We add noise to p(x|z) in a specific way to ensure numerical stability of the mutual information estimation.
> >
> > In the main paper, this may be confusing because in the next paragraph, we further introduce how we constrain the marginal in p(z) and its implications. To clarify these points, we rewrote these paragraphs as below:
> >
> > The mutual information estimator also requires the prior of $p(Z)$ to have a factorized marginal, $\mathcal{N}(0,I)$. This is a typical constraint introduced in the original variational autoencoder \cite{Kingma2013}. Combining this factorized marginal constraint on $p(Z)$ and the above constraint on $p(X|Z)$, we can approximate $I(X;Z)$ with the following estimator,
> >
> > Using $p(Z)$ with a factorized Gaussian marginal has other implications: it is the sufficient and necessary condition for the latent space to exhibit orthogonal symmetry \cite{Skitovitch1953, Darmois1953, Lukacs1954}. This Darmois-Skitovitch characterization was first introduced to identify unique factors for independent component analysis \cite{Hyvrinen1999,Peters2017}. Recent work also showed that this factorized marginal $Z$ is necessary to capture “ground truth factors of variation” \cite{Bengio2013} in the latent representation $Z$ \cite{Higgins2017, Kumar2020}.
> >
> >
> > At various places, I lost track which analysis were based on the U-Net and which ones were based on the additional VGG.
> > Throughout the paper, we use the feature space from a pretrained VGG19 to represent frames from natural movies (instead of using pixels). We rewrote the encoder-decoder part to clarify this:
> >
> > Because a movie frame contains features of multiple scales, we use the perceptual loss \cite{johnson2016} as the loss function. This loss function compares a reconstructed frame with its target frame with respect to learned features from a pretrained VGG, as opposed to their raw pixels. Throughout the paper, we use features from pretrained VGG19 to represent movie frames.
> >
> > The training section in the supplementary material suggests that different samples were generated by dropping 2 cells randomly, but there is no explanation how the network could handle input with permuted cell identities sensibly.
> > We did not permute our cells when we randomly drop out two firing cells (set them to zero). We will change our supplementary material to clarify this.

---

> > > ### Comment · Reviewer_YUEv · 2022-08-09
> > > **Acknowledging Author Response**
> > >
> > > As the other reviewers are surprisingly positive for this submission, I read through the discussions here carefully again. Unfortunately the responses here did not convince me that the methods the authors propose are actually helpful though.
> > >
> > > In particular, I know the theory of variational auto encoders of course. I may have written a bit flippant in my original review, but this is exactly what I refer to with the mostly unjustified noise additions in hope of finding interpretable features.
> > > The theory of variational auto-encoders is well suited for illustrating my main point here though: In information theoretic terms the frames in a 9 second movie at 24 Hz have about 1 Byte of entropy (log2(9*24) = 7.754 bit) and I am sure all frames are distinguishable based on the neural responses. Thus, it is not surprising that time or frame ID are decodable from the bottleneck as this is the essentially all information there is.
> > >
> > > The authors interpret their observations as an insight about retinal coding, but I strongly disagree. Given the extremely small stimulus set used here, there is no way of distinguishing the many different features that are completely correlated in the data.
> > > Using a large deep neural network does not remove the need to show that the encoded information is not caused by alternative explanations and in fact most of the work the authors cite here is actually towards excluding such alternatives. In the specific case discussed here, I am still almost perfectly certain that finding the temporal information is not an insight into retinal coding but is due to some artifact due to the stimulus presentation and representation in the network. The analyses the authors added during the rebuttal phase actually increased my believe in this explanation not decrease it.
> > >
> > > Thus, I overall keep my clear rejection rating.

---

> > > > ### Author Response · Authors · 2022-08-09
> > > > **Thank you for the update. Unfortunately, we find some comments/criticism are not relevant/impossible within biological limit.**
> > > >
> > > > Thanks for reading our reply. We apologize that there are still some misunderstanding on the primary aim and the neuroscience context of our paper.  Unfortunately, we also find some of the comments not relevant for our paper or impossible within the current experiment limit for us to improve our work.
> > > >
> > > > 1) "a 9 second movie at 24 Hz": we clearly stated that our movies are 20s long with 60 frame per second. We do not understand why the reviewer cites this entropy corresponding to "a 9 second movie at 24 Hz": as the reference of "all information there is."
> > > >
> > > > 2) "the extremely small stimulus set": As we stated in our initial reply, there are real, biological constraints on how long the stimuli can be. Retinal computations are generally carried out within a time range of 100–400 ms, according to experiments employing the retina of salamanders, mice, cats, and other animals. Natural stimuli lasting 20 seconds are sufficiently long to obtain large varieties of retinal responses. Visual coding happens in fast timescales. The Allen Brain Observatory and a recent study (Xia et al., 2021) also use natural movies with similar durations, such as 30s, to capture the responses of the visual cortex (a neural system that may act at longer timescales than the retina) to natural movies. Even so, they have fewer movies (1-2 movies) and fewer trials every session (10-30 trials vs. 90 trials used here). We have a dataset that is the comparable other publicly available datasets for visual coding. Getting a larger dataset would require an order of magnitude improvement in recording sessions which is currently not possible.
> > > >
> > > > Xia et al., 2021 Stable representation of a naturalistic movie emerges from episodic activity with gain variability
> > > >
> > > > 3) "there is no way of distinguishing the many different features that are completely correlated in the data."
> > > > In Ganmor et al., 2011, they showed that a 20-cell population can encode up to 2 bits as noise correlation in their instantaneous (17ms, 1 time bin) firing pattern. Here, we used 500ms (30 time bins) and 45-cell population, it is plausible that such a large population with long duration can encode frame ID/time for 100 frames (log2(100) = 6.6 bits).
> > > >
> > > > Ganmor et al., A thesaurus for a neural population code 2011
> > > >
> > > > 4) As we explained in the main paper, we froze the weights of trained model prior to obtaining latent activations using hold-out test sets. Data processing inequality applies here: the frozen model is a stochastic function of the input retinal code. All information the latent activations encode about the stimuli must come from its input retinal code.
> > > >
> > > > 5) "mostly unjustified noise additions in hope of finding interpretable features." We had made it clear in our initial response that we added noise in order to prevent posterior collapse and facilitate mutual information estimation, not directly imposing interpretability. This is a method used (Kolchinsky, 2017, Saxe et al. 2018) to facilitate mutual information estimation as well. For your convenience, we attached our session for adding noise below. We do not use adding noise to justify finding interpretable features at all:
> > > >
> > > > As discussed in \cite{Razavi2019,Kumar2020, Locatello2019}, we can obtain a meaningful $Z$ by making the posterior to have a small but nonzero noise. To be specific, we have $\sigma^2>0$ for $p(X|Z)$. This is also dubbed a 'committed rate' for the encoder. In our case, we choose $\log\sigma^2=-1.0$ to further ensure numerical stability of the mutual information estimator that we use \cite{Kolchinsky2017}.
> > > >
> > > > The mutual information estimator also requires the prior of $p(Z)$ to have a factorized marginal (each marginal is an independent Gaussian), $\mathcal{N}(0,I)$ (they are independent Gaussians). This is a typical constraint introduced in the original variational autoencoder \cite{Kingma2013}. Combining this constraint on $p(Z)$ and the above constraint on $p(X|Z)$, we can approximate $I(X;Z)$ with the following estimator,
> > > >
> > > > Kolchinsky, 2017 Nonlinear information bottleneck
> > > > Saxe et al., 2018 On the Information Bottleneck Theory of Deep Learning

---

### Official Review · Reviewer_vaQu · 2022-07-08

**Rating:** 7
**Confidence:** 3
**Soundness:** 3 good
**Presentation:** 3 good
**Contribution:** 3 good

**Summary:**


In this manuscript the authors describe the use of a deep-learning pipeline to understand the nature of encoding in the salamander retina. Namely, they collect PSTHs from neurons in response to short natural movies, then the PSTHs are converted to images and used as the input to a U-Net, which is trained to reconstruct the movie. Curiously, the latent representation can be used to accurately decode time in the movie, even for completely novel movies. This suggests that the model learns a low-dimensional representation of elapsed time. Lastly, the authors use information theoretic analysis to show that encoding of time requires jointly encoding of static and dynamic features.


**Questions:**

1. Did the authors attempt to use simpler models to decode time?
1. Can you predict time in a held-out movie using visual features from another? Might be a remote possibility, but I want to know if there are any "simple" visual features that predict time. If you take frames from the fish movie and compute the pairwise distance to all frames in another movie, I assume there is no correlation with time?
1. Did the authors try simple autoencoder architectures with a single latent layer rather than a U-Net? I am curious what benefit is conferred by the multi-resolution architecture, especially given that they are using a GAF representation of spike trains.

**Limitations:**

I found the discussion of limitations adequate. No concerns with respect to societal impact.

**Strengths And Weaknesses:**

Strengths

1)	I found the pipeline presented by the authors to be fairly novel and well-motivated. I am not aware of the use of Gramian angular fields and autoencoders to understand population coding. This has the potential to be useful in other fields of neuroscience more generally.

2)	Their result – that the salamander retina may have a movie-invariant code for time – is genuinely interesting.

3)	For the most part the paper was well-written.

Weaknesses

1)	Lack of controls and comparisons with other methods. I do not know how likely this is, but one could imagine that time could be encoded by fairly trivial aspects of the data – e.g. a change in the aggregate statistics of spiking. Perhaps neurons fire more to the onset of the movie and slowly decay to 0 by the end. Since simple controls aren’t presented, it is hard to know if this sort of triviality explains their results.

    1.	First, the authors could simply shuffle the identity of neurons in the test phase. This would account for any trivial gross changes in spiking statistics accounting for decoding accuracy.

    1.	Second, a much simpler model should be presented as a baseline. For instance, train a linear regression model to predict the time in the movie from the PSTHs (or a dimensionally reduced version of them), in addition to summary statistics computed across neurons (e.g. the mean and variance). How does the accuracy of this baseline model compare to the U-Net? The outcome of this comparison would provide a much better motivation for the authors’ pipeline. While it is novel, it is important to know how much extra information it provides.

    1.	Third, show more raw results. Visualize how the latent space evolves over time for each of the three movies along with the PSTHs. It would be nice to have a better intuition for how the model is transforming the spikes, and what features in the spike trains are relevant for encoding time. Perhaps there are no obvious features one can spot by eye, which provides even better motivation for the authors’ method.


Minor

1)	The authors mention domain-conversion as an application of U-Nets. This could be my own misunderstanding, but I am not sure what is meant here since one of the examples given is super-resolution imaging.

2)	Figure 2 caption has bottom row twice. I also don’t know how the authors computed this receptive field. More detail would be nice here.

---

> ### Author Response · Authors · 2022-08-02
> **Thank you for your detailed feedback. We included a separate section and 3 videos/6 figures/5 simple decoder methods in the supplementary material**
>
> Thank you very much for these detailed comments. We agree that it is important to compare our results with other models to establish baseline expectations for the difficulty of “time in natural scene” from retinal recordings. We add a separate section in the supplementary pdf to address your concerns (We cannot post them here because of the character limit). Overall, we show that “time in natural scene” is not a feature that is trivially encoded.
>
> Did the authors try simple autoencoder architectures with a single latent layer rather than a U-Net? I am curious what benefit is conferred by the multi-resolution architecture, especially given that they are using a GAF representation of spike trains.
>
> We use the GAF representations for its engineering benefit. This enables us to initialize the encoder portion of U-net with pretrained weights from ResNet18. Initializing some portion of a deep neural network model with structured weights (instead of randomization) speeds up its training. It also helps our model to escape early local sub-optimal solutions. Recent work in financial forecasting (Barra et al., 2020) shows that the Gramian angular field not only captures subtle statistical structures of time series, but the engineering advantage from using a 2D image (initialization with pretrained ResNet18) is impressive. In our case, the decoder portion is always initialized with random weights.
> Barra et al., Deep Learning and Time Series-to-Image Encoding for Financial Forecasting 2020
>
> We use U-net to generate realistically looking natural movie frames. Image samples generated from a trained VAE tend to be blurry (Goodfellow et al., 2014; Larsen et al., 2015). Previous works show that this is because maximum likelihood- type loss functions for VAE primarily penalize putting a feature in the wrong spot, much more than they penalize blurring out or entirely skipping over that feature. The U-net here circumvents this issue by using multiple variational sampling layers responsible for decoding features from coarse to fine scales. These variational layers are trained to match features at different spatiotemporal scales of natural movies separately. Therefore, a feature that gets blurred out at a broad scale (and thus ignored by the loss) may be penalized as misplaced features at a fine scale (and thus corrected at a fine scale). When U-net generates a movie frame, it uses decoded features from a fine scale to “sharpen” blurry features of a broad scale. This helps us to generate movie segments very similar to the ground truth (shown in supplementary material). Other works using hierarchical VAE also observe a similar benefit [*]
> Goodfellow et al., Generative Adversarial Nets. NeurIPS 2014
> Larsen et al., Autoencoding beyond pixels using a learned similarity metric. 2015
> Maaløe et al., BIVA: A very deep hierarchy of latent variables for generative modeling. NeurIPS 2019
> Vahdat et al., NVAE: A Deep Hierarchical Variational Autoencoder. NeurIPS 2020
>
> Second, we find that disentangled features from different variational sampling layers show different spatio-temporal scales. In the main paper, we found the background/motion separation within a broad-scale variational sampling layer. We also find other disentangled features in a different variational sampling layer at a finer scale. In the example below, the left feature below only encodes the find scale stripes on the zebrafish body when the fish is in the top part of the movie frame.
> https://imgur.com/a/4BlMuqG
> The disentanglement between these two features occurs only when a fish swims through the bottom half of a movie frame. It is more subtle because it is only visible during a short duration within the fish movie. Combining the disentangled features at broad scale (background/motion) and fine scale (part of a scene), our future work will focus on discovering features more complex than time in the natural scene.
>
> Minor
> The authors mention domain-conversion as an application of U-Nets. This could be my own misunderstanding, but I am not sure what is meant here since one of the examples given is super-resolution imaging.
>
> Thanks for pointing out this confusion. We changed the example to text-to-speech [Unet-TTS]. We would like to clarify that variational sampling adds extra flexibility to learn consistent features across different modalities (e.g, estimating 3D joint-angle configurations of human hands from 2D RGB images) in the final version (now subject to the 9 page limit)
>
> Figure 2 caption has bottom row twice. I also don’t know how the authors computed this receptive field. More detail would be nice here.
> We corrected the Figure 2 caption by referring to them as: 1st row (from top to bottom), 2nd row, 3rd row and 4th row. We compute the receptive field by first converting the reconstructed feature frame into its aggregated x-axis activation (by summing along the y axis), then we concatenate all the aggregated x-axis activations along the time axis.

---

> > ### Author Response · Authors · 2022-08-04
> > **We attached our simple decoder and visualization results here for your convenience**
> >
> > We just realized that NeurIPS2022 allows multiple comments. We included our simple decoder methods here for your convenience (all these results are also in the supplementary material).
> >
> > Simple visualization I: mean PSTH of the retinal population.
> > Here, we show the mean PSTH of the entire retinal population (the gray region is the standard error per time bin). They do not show a clear trend that correlates with time (e.g., neurons that fire more at the onset of the movie and slowly decay to 0 by the end.)
> > https://imgur.com/a/YXcbD45
> > https://imgur.com/a/wYMQKvW
> > https://imgur.com/a/pcDt7dX
> >
> > We also show the 400ms PSTH patterns from 5 example neurons responding to the fish movie. Their individual spiking patterns also do not correlate in any simple way with time since movie onset.
> >
> > https://imgur.com/a/7wqSzBu
> >
> > Simple visualization II: Frame-to-frame distance between different movies
> >
> > Here we show the pairwise frame-to-frame distance between different movies. This follows the reviewer’s advice to “take frames from the fish movie and compute the pairwise distance to all frames in another movie.” We hypothesize that if some trivial visual features of one movie can encode time of a different movie, then there may be a correlation between frame-to-frame distance and time when both frames correspond to the same time in both movies. We use the same frame-to-frame distance in Fig 4 of the main paper. Similar to mean PSTHs, we do not observe a clear trend that correlates with time.
> > Fish vs. leaf:
> > https://imgur.com/a/0CFApfr
> > Fish vs. water:
> > https://imgur.com/a/kliSyvJ
> >
> > Simple visualization III: 2D dimensionality reduction of the latent space
> > Although a linear decoder can decode time from a 10D latent representation, time is not encoded by any easily discernible trivial aspect of the latent space. When we convert the 10D latent activations into 2D with Isomap, we do not observe any obvious correlation between how latent activations change through time and time itself.
> >
> > fish: https://imgur.com/a/4cIynLl
> > leaf: https://imgur.com/a/bSOxMvR
> > water: https://imgur.com/a/QSidiiM water
> >
> > Linear decoder (raw/simple dimensionality reduction)
> >
> > |           |Instantaneous PSTH’ |Raw 500ms PSTH (linear decoder) |Shuffled PSTH | Isomap (d=10) | PCA (d=10)|
> > |--------|-------------------------|----------------------------------------|------------------|------------------|--------------|
> > |Fish    | 3.9%                          | 99%                                                | 60.2%             | 14.9%              | 55.9%        |
> > |Leaf    |4.5%                           | 70%                                                | 5.6%               | 7.0%                | 2.5%          |
> > |Water | 4.7%                           | 99%                                                | 65.9%             | 16.6%              | 32.1%        |
> >
> >
> > We show here the performance (percentage correct) of 3 different linear decoders using raw PSTH’s to decode time. A linear decoder trained on shuffled PSTH’s (shuffled neuron identity per input) shows inferior performance compared to one trained on raw PSTH's. This means that decoding time does not come from trivial gross changes in spiking statistics as time passes during the movie. All these raw PSTH’s are high-dimensional: 45 cells X 30 time bins (17ms each) = 1350 dimensions. A linear decoder trained on these high-dimensional raw PSTH’s works well for two movies (fish and water), but much worse for the leaf movie. It is possible that there are different nonlinear components in the retina code responding to different movies. This shows a clear advantage of U-net: a linear decoder trained on the U-net’s latent representation with only 10 dimensions  can decode time in the natural scene with 99% accuracy for all three movies (Fig 3).
> >
> > Next we investigate whether simple dimensionality reduction methods can also find low-dimensional, generalizable features for the time in the natural scene. We use two off-the-shelf dimensionality reduction methods to obtain low-dimensional representations of raw PSTH’s: One is ISOMAP. It is a nonlinear dimensionality reduction method. We choose ISOMAP over another popular option (tSNE) because ISOMAP allows us to train an embedding space using the fish movie onto which we can project the other two movies. The other is PCA. It is linear. We choose d=10 because the U-net-learned latent representation achieves a 99% decoding performance with d=10. We find that neither methods show decoding performance that are comparable to what we achieve using the latent representation from a trained U-net. It is surprising how bad the ISOMAP is. Using PCA, the decoding performance varies significantly across different movies. Echoing from the decoding result using raw PSTH’s, this suggests that retinal activity responding to these natural movies have complex and diverse structures. This diversity makes it challenging to discover the generalizable features across different movies.

---

> > > ### Comment · Reviewer_vaQu · 2022-08-05
> > > **Thank you for the detailed revision!**
> > >
> > > I appreciate the extra analysis here, which clarifies a a lot. It is reassuring that the authors cannot get results as good as the U-Net with many of the controls. However, one thing that is concerning about the new results is that the shuffled PSTH can nearly outperform 10D PCA. What's going on here? I would make sure that this is made very clear in the discussion, as this suggests that a large part of the variance (~60%) in at least two of the movies *does* in fact come from some trivial trends in the data. At this point I won't suggest additional analysis due to time constraints, but this should be presented as a clear caveat.  Thankfully the U-Net does a good job with the leaf movie, which can't be explained by this issue.

---

> > > > ### Author Response · Authors · 2022-08-09
> > > > **Thank you very much for raising this concern. We add a few back of the envelope estimations for the trivial aspects of the fish movie**
> > > >
> > > >
> > > > one thing that is concerning about the new results is that the shuffled PSTH can nearly outperform 10D PCA. What's going on here?
> > > >
> > > > TL;DR: We quantify that there are 17.8%-31.7% trivial trends in the fish movie. Because the latent feature space learned by our variational U-net is generalizable across all three natural movies, this latent space does not rely on such trivial aspects of the data to decode time.
> > > >
> > > > We appreciate the reviewer bringing up this issue. We would like to look into the reasons behind why the shuffled PSTHs and 10D PCA demonstrate comparable decoding accuracy. This will give quantifiable information about the amount of trivial trends in the data. We postulate that the statistical structures that are constant or present in both the shuffled PSTH's and the raw PSTH ( (or its compressed 10D PCA) are the trivial elements of the data. In the narrow context of training linear decoders, we understand these trivial aspects as linear features that are informative for decoding the time in the natural scene linearly. If linear decoders on both shuffled PSTH’s and 10D PCA use these shared trivial aspects to decode frame numbers, both linear decoders should make similar predictions and similar errors, i.e., they may concur on a) how they predict individual samples and b) how they make prediction errors. We begin by determining whether these two linear decoders are similar with regard to a) and b).
> > > >
> > > > 1) We would like to understand whether such a similar decoding accuracy corresponds to similar predictions, i.e., for a given sample, do the shuffled PSTH and 10D PCA agree on their predictions? We discover that only 31.7% of the test samples for the fish movie are correctly predicted by both the shuffled PSTH and 10D PCA. This percentage reduces to 20.3% in the water movie. Therefore, there may be linear features that are consistent between the shuffled PSTH’s and 10D PCA of the fish movie, but they are not as much as 60%. Our variational U-net does not leverage these insignificant patterns to attain 99 percent accuracy across all three natural movies because the shuffled PSTH's struggle to decode the leaf video (showing just a 5.6% accuracy in our previous reply).
> > > >
> > > >
> > > > 2) We also demonstrate that the prediction errors produced by the shuffled PSTHs and the 10D PCA linear decoders are different. This is just another example of how the 10D PCA and the shuffled PSTHs differ from one another.
> > > >
> > > > We show below the mean errors for both fish and water movies. The x-axis of these plots are ground truth frame numbers (1-100). The y-axis is the mean difference between predicted frame labels and their ground truth frame labels. We notice that the mean errors produced by the 10D PCA are within (-5,5), indicating that 10D PCA maintains a rough temporal order.  However, the shuffled PSTH’s show more violations of the temporal order of the test frame labels. Here, mislabeling an earlier frame as a much later frame number corresponds to a positive error (e.g., frame 6 predicted as frame 44 yields an error of 38). Mislabeling a later frame as an earlier frame corresponds to a negative error (e.g., frame 70 predicted as frame 32 yields an error of -32).
> > > >
> > > > Fish movie: https://imgur.com/a/NSbz2hf
> > > >
> > > > Water movie:  https://imgur.com/a/r1QZIWK

---

> > > > > ### Author Response · Authors · 2022-08-09
> > > > > **(continued) back of the envelope estimations for the trivial aspects of the fish movie**
> > > > >
> > > > > Because linear decoders trained on the shuffled PSTH’s and 10D PCA are different, there are a few possibilities: 1) the shuffled PSTH’s contains principle components beyond the 10D PCA (10D PCA explains about 51% of the explainable variance in the raw PSTH’s); 2) The shuffled PSTHs and 10D PCA have drastically different dimensionalities: 1350 vs. 10. With 99% accuracy, our variational U-net identifies a low-d (d=10) feature space that can be applied to all three movies. In the high-dimensional (d=1350) neural space, the raw PSTH's corresponding to various frame labels may group into clusters that are sufficiently separated from one another. This “far enough” may enable these clusters to be linearly separable in the high-d space even after they are shuffled, albeit differently from a linear decoder trained on the raw PSTH’s. We hypothesize that a linear decoder trained on shuffled PSTH’s is different because it uniquely shows broad range violations of temporal order. If this is the case, the shuffled PSTH’s may not share linear features with the raw PSTH’s. We investigate this by first projecting the shuffled PSTH’s to the 10-50 principal components of the raw PSTH’s and then train linear decoders on the transformed shuffled PSTH’s.
> > > > >
> > > > > 3) We observe that linear decoders trained on the transformed shuffled PSTH's exhibit poor decoding accuracy when we project the shuffled PSTH's to the first 10 to 50 principal components obtained from the raw PSTH's.  We stop at the 50D PCA because it achieves 97.2% accuracy, i.e., adding additional linear features from the raw PSTH’s wont help much to decode time in the natural scenes.
> > > > >
> > > > > |Fish movie                                 | d=10    | d=15    |  d=20   | d=30    | d=40     | d=50     |
> > > > > |------------------------------------|--------- |----------|-------- |--------- |----------|-----------|
> > > > > |PCA(raw)                                   | 55.8%  | 60.7%  | 80.2% | 93.6%   | 95.3%   | 97.2%   |
> > > > > |Transformed shuffled PSTH’s    | 9.3%   | 12.9%   | 13.8% | 14.9%   | 16.1%   | 17.8%  |
> > > > >
> > > > >
> > > > > |Water movie                              | d=10    | d=15    |  d=20   | d=30    | d=40     | d=50      |
> > > > > |------------------------------------|---------|----------|---------|----------|----------|-----------|
> > > > > |PCA(raw)                                   | 32.1%  | 42.8%  | 55.4%  | 72.7%  | 80.4%  | 85.2%   |
> > > > > |Transformed shuffled PSTH’s   | 5.7%   | 7.5%     | 9.6%    |10.5%   | 11.8%  |13.4%    |
> > > > >
> > > > > This analysis shows that the raw PSTH’s and the shuffled PSTH’s do not share linear features that are useful for decoding time in the natural scene. Only 9.3%-17.8% accuracy is obtained once the shuffled PSTHs are converted to 10D-50D PCA. All these are much lower than the shared accuracy (31.7%) between 1350-d shuffled PSTH’s and 10D PCA. Considering that we only empirically observe prediction errors that violate temporal order using the shuffled PSTH’s, it is possible that in the 1350-d space, a linear decoder trained on the the shuffled PSTH’s may find structures that are linearly separable but are not used by the raw PSTH’s to decode time. Because a linear decoder trained on 50D PCA can achieve 97.2% accuracy, this 50D-PCA includes almost all linear features in the raw PSTH’s that can be used to decode time in the natural scenes. Since the shuffled PSTH’s transformed to the 50D space can only obtain a 17.8% accuracy, this may be a “lower bound” of how much trivial trends are present in the fish movie.
> > > > >
> > > > > These PCA calculations also suggest our variational U-net carries out a substantial amount of nonlinear transformation to learn a feature space with a much lower dimensionality (d=10) that is generalizable across all three natural movies. If our submission is successful, we will add clarification to the decoding section and supplementary material.

---

> > > > > > ### Comment · Reviewer_vaQu · 2022-08-09
> > > > > > **Thanks again for really digging in**
> > > > > >
> > > > > > All of my concerns are addressed, so increasing my score to reflect the additional effort here.  Thanks again to the authors.

---

> > > > > > > ### Author Response · Authors · 2022-08-09
> > > > > > > **Thank you for the positive feedback**
> > > > > > >
> > > > > > > Thank you for your positive response and for updating the score. We are glad these points addressed your concerns and we will make sure to add them in the revised version of the paper

---

### Official Review · Reviewer_VKvw · 2022-07-09

**Rating:** 7
**Confidence:** 4
**Soundness:** 4 excellent
**Presentation:** 3 good
**Contribution:** 3 good

**Summary:**

In their paper "Learning low-dimensional generalizable natural features from retina using a U-net", the authors modify a U-net with a ResNet encoder and variational sampling layers for skip connections to identify salamander retinal population response space-time features. They do this by reconstructing movie frames from neural responses. Moreover, they analyze the latent representations and find that static features and dynamic features are encoded synergistically.

**Questions:**

Do the authors see other potential applications of their U-net modifications outside of neuroscience?

Figure 2 caption: Bottom row appears twice. The authors should properly refer to all four rows.

Figure 3: The authors trained on the fish movie and tested on the other two movies. Would the results be similar if training on the leaf or water movie and testing on the other two movies?

Line 248: Network misspelled.

Eq. 2: Please add brackets for clarity. Moreover, there is no explanation that this approximation comes from an independent Gaussian assumption. Please clarify the text.

Line 270: Subscript "joint" broken.

**Limitations:**

The authors do not discuss any negative societal impacts of their work. In my opinion, this is adequate in this case.

There is an independence assumption, which is very briefly discussed in the Discussion. The authors also discuss the lack of proper predictive constraints and the particular type of synergy that they looked at. Overall, limitations have been appropriately discussed.

**Strengths And Weaknesses:**

The modifications of U-net in terms of the ResNet encoder and variational sampling layers for skip connections is an interesting innovation that might have broader applications and could be of interest to the wider machine learning community. However, the authors do not discuss other potential applications of their methodology and instead just mention that the architecture can learn many compressed latent representations simultaneously.

To the best of my knowledge, the finding that static features and dynamic features are encoded synergistically in the retina is novel. Given that this finding helps to elucidate how the retina processes visual stimuli, this is a significant contribution. Therefore, the paper should be of interest to the wider neuroscience community.

The paper is written in a clear and compact way except for a couple of minor points, which I mention in Questions. Figures are informative and are well integrated into the text.

I could not find any conceptual or technical mistakes. The authors clearly state their assumptions and analyze their data rigorously.

EDIT: The authors addressed my minor concerns. I increased my rating to 7.

---

> ### Author Response · Authors · 2022-08-02
> **Thank you for your detailed feedback, we updated the main paper accordingly and did additional calculations to address your questions**
>
> Do the authors see other potential applications of their U-net modifications outside of neuroscience?
>
> Yes, we updated the last paragraph of our paper to include a few potential applications outside of neuroscience. If this submission is successful, we will use a separate paragraph to include more details in potential applications outside of neuroscience. We attached the paragraph below for your convenience:
>
> Compared to other methods that learn a latent representation between neural activity and external stimuli, our work is the most similar to \cite{Liu2021, Zhou2020}. They also used a highly expressive, feedforward encoder (a multilayer perceptron (MLP)). MLP is fully-connected, so that its learned latent representation corresponds to a single global scale. Unlike the MLP, our U-net architecture uses a ResNet as the encoder.  The ResNet encoder attains the same performance as the MLP, but by cascading Resblocks from coarse-to-fine scales. This enables the U-net architecture to learn compressed latent representation at multiple scales simultaneously. We did not explore this feature explicitly, but it may be interesting for future work in understanding neural dynamics in complex natural environments. For example, there is a hierarchy of timescales both in natural scenes and output behaviors, from hundreds of milliseconds to minutes (whisking to walking to making action plans \cite{Recanatesi2022,Stern2021}). With additional constraints \cite{Khemakhem2019}, variational sampling layers may hierarchically learn interpretable latent representations for each timescale, individually, and understand how they can be combined to drive complex behavioral outputs. Outside of neuroscience, This variational U-net is compatible to learn latent representations between other temporal sequences (e.g., text) and complex spatio-temporal signals (speech or video). Two potential applications are text-to-speech and video summarization. Combining latent representation at multiple scales may also reveal semantic relationships between complex features in general object recognition, e.g, how does a model combine local features (nose, eye) with global shape (e.g., body size) to discriminate between cats and dogs.
>
> Figure 3: The authors trained on the fish movie and tested on the other two movies. Would the results be similar if training on the leaf or water movie and testing on the other two movies?
>
> Yes, when we train on the leaf/water movie and test on the other two movies, our decoding performance (percentage correct) are similar. Please find the results below:
>
> |             | Water (d=5) | Water (d=10) | Leaf (d=5) | Leaf (d=10)|
> |-----------|------------------|-------------------|---------------|----------------|
> |    Fish   | 78.2%         | 96.9%            |    72.2%    |  98.0%       |
> |   Leaf    | 79.5%         | 97.8%            |   84.4%    |  99.1%        |
> |  Water  | 84.0%         | 97.4%            |    71.7%    | 97.9%        |
>
> Minor comments:
> Figure 2 caption: Bottom row appears twice. The authors should properly refer to all four rows.
> Line 248: Network misspelled.
> Eq. 2: Please add brackets for clarity. Moreover, there is no explanation that this approximation comes from an independent Gaussian assumption. Please clarify the text.
> Line 270: Subscript "joint" broken.
> We corrected these typos. We also rewrote the paragraph before the MI estimation to clarify the assumption on P(Z) and P(X|Z).
>
> The mutual information estimator also requires the prior of $p(Z)$ to have a factorized marginal (each marginal is an independent Gaussian), $\mathcal{N}(0,I)$ (they are independent Gaussians). This is a typical constraint introduced in the original variational autoencoder \cite{Kingma2013}. Combining this constraint on $p(Z)$ and the above constraint on $p(X|Z)$, we can approximate $I(X;Z)$ with the following estimator,

---

> > ### Comment · Reviewer_VKvw · 2022-08-09
> > **Rating increased**
> >
> > I thank the authors for their revision and for correcting minor mistakes that I pointed out. The updated paragraph is useful and describes sensible potential applications outside of neuroscience. Moreover, the results from training on the leaf and water movies and testing on the others are reassuring. Therefore, I increased my rating from 6 to 7.

---

> > > ### Author Response · Authors · 2022-08-09
> > > **Thank you for the update**
> > >
> > > Thank you for your positive feedback and for updating the score. We are glad these points addressed your concerns and we will make sure to include the decoding with the other two movies in the revised version of the paper.

---

### Official Review · Reviewer_PGRz · 2022-07-11

**Rating:** 7
**Confidence:** 3
**Soundness:** 3 good
**Presentation:** 3 good
**Contribution:** 2 fair

**Summary:**

The authors adopted a task-agnostic U-net to model the retinal encoding process and characterize the representation of "time in the natural scene" in a low-dimensional compressed latent space. The U-net takes PSTHs of the salamander retinal population as input and the reconstructed movie frames as output, and learns with error propagation. They found that the retina has a generalizable encoding for time in the natural scene, which learned from one movie can be used to represent time in another movie. They also showed that static textures and velocity features of a natural movie are synergistic.

**Questions:**

Q1: since neural responses are delayed (when an input stimulus is presented), how to align the input (neural responses) and the output (movie frames) to make a proper input-output pairs? Is it necessary to shift few miniseconds to find a good alignment?

Q2: beside the background and motino seperation showed in Fig2, is there any other disentangled feature found in the decoder layer? Is this feature disentanglement due to the utilization of the variational sampling layers? Any other tricks to find these disentangled feature?

Q3: does the decoding performance reported in Fig3 means to decode the temporal order of N movie frames?

**Limitations:**

Yes, the authors addressed the limitations.

**Strengths And Weaknesses:**

The paper is clearly organized and the logic flow is good. Discovering the low-dimensional feature representation in retinal neuronal firing (the time feature) with the combination of a ML based method is interesting.

---

> ### Author Response · Authors · 2022-08-02
> **Thank you very much for your feedback and questions**
>
> Q1: since neural responses are delayed (when an input stimulus is presented), how to align the input (neural responses) and the output (movie frames) to make a proper input-output pairs? Is it necessary to shift few miniseconds to find a good alignment?
>
> We agree with the reviewer that the neural responses have an intrinsic delay to flashed stimuli (50-70ms). We do not shift because all results here are based on how a large retinal population performs bona fide prediction. Previous work [Palmer et al., 2015] showed that a small retinal population (9 cells) can already predict 150ms into the future using its instantaneous spikes when the input is a correlated temporal sequence. Here we use neural responses of a much larger retinal population and an extended PSTH history (t, t+500ms) to predict an unseen future frame at t+600ms. Including the intrinsic delay, this setting lets us predict 150 -170ms ms into the future. This range is biologically plausible.
>
> Palmer et al., predictive information in a sensory population, PNAS 2015
>
> Q2: beside the background and motion separation shown in Fig2, is there any other disentangled feature found in the decoder layer? Is this feature disentanglement due to the utilization of the variational sampling layers? Any other tricks to find these disentangled feature?
>
> 1) We show the background and motion separation in the main paper because these resemble previous experimental findings in the retina. We also observe other disentangled features. For example, the left feature below only encodes the fine-scale stripes on the zebrafish body when the fish is in the top part of the movie frame. Its disentanglement with the right feature is transient (i.e., when a fish only swims in the bottom part) and specific to this particular movie segment.
> https://imgur.com/a/4BlMuqG
> 2) We find that the variational sampling layer at this specific decoding layer generates these disentangled features. To show this, we include two examples below: one with this specific latent activations and one without. We find that removing the latent activations silences both background and motion features. (They are also in the updated supplementary zip file, you can find them as rebuttal_figures/feature_output_before/after.gif)
> i) with latent representation
> https://imgur.com/a/RNb1NnH
> ii) without latent representation (these are whole frame flashes without spatial features)
> https://imgur.com/a/jcN4oDL
>
> 3) Adding additional disentanglement constraints is an interesting future direction. In general, finding disentangled representation of a real-world dataset is an active research subject. Previously successful examples focused on generating representations that match a pre-existing semantic understanding of data. One prominent example is the $\beta$-VAE (Higgins et al., 2016). It uses adjustable hyperparameters to discover factors that act as a part based representation of chairs (e.g, chair legs, sitting areas, etc) or human faces (eye, nose, mouth, etc). This particular work helped us to generate a first pass semantic understanding (based on static/dynamic features) of natural movies. We plan to pursue disentangled features in our future work.
>
> Higgins et al., beta-VAE: Learning Basic Visual Concepts with a Constrained Variational Framework ICLR2016
>
> Q3: does the decoding performance reported in Fig3 means to decode the temporal order of N movie frames?
>
> In this particular work, we obtain the temporal order of N movie frames because the neural responses have their own intrinsic temporal structure, and we take a temporally extended slice (500ms) of those data. It would be a harder challenge to decode long sequences of frames using instantaneous spiking patterns without overlap. This is an interesting future direction.

---

> > ### Comment · Reviewer_PGRz · 2022-08-09
> > **Thank you for the response**
> >
> > The authors answered most of my concerns and questions. Would love to hear from other reviewers during the Reviewer- Metareviewer Discussion.

---

> > > ### Author Response · Authors · 2022-08-09
> > > **Thank you for the update**
> > >
> > > Thanks for your continued positive response of our work. We are glad these points addressed your questions and we will definitely include more clarification on the range of prediction and the disentanglement between features in the revised version of the paper.

---

### Author Response · Authors · 2022-08-09
**General Response**

We would like to express our gratitude to all reviewers for their insightful comments and recommendations.  We provided individual responses for each review below. We are very grateful for the favorable comments that described our pipeline as an interesting innovation (VKvw), novel and having the potential to be useful in other fields of neuroscience (vaQu). We also appreciate the positive feedback on our neuroscience findings which described the low-dimensional, movie-invariant representation of time as interesting (PGRz, vaQu), and the finding that static features and dynamic features are encoded synergistically as a significant contribution (Vkvw). During the rebuttal period, we added results showing that our variational U-net learns a low-dimensional movie-invariant representation of time when it is trained on the other two movies. We also demonstrated how this generalizable, low-dimensional feature space differs significantly from linear or other off-the-shelf dimensionality reduction methods. We look forward to incorporating these suggestions in a revised version of the paper.

---

### Meta-Review · Area_Chair_9HXJ · 2022-08-27

**Recommendation:** Accept
**Confidence:** Less certain

**Metareview:**

The authors analyze the latent representation of visual features from natural movies in the salamander retina using a U-net. They train an encoder to learn a compressed latent representation from a large population of salamander retinal ganglion cells responding to natural movies, while a decoder samples from this compressed latent space to generate the appropriate movie frame. They characterize its representation of “time in the natural scene” in the latent space of their model.

Overall, the reviewers expressed a lot of interest in the topic and valued the novel application to salamander retinal data. There were some questions about the significance of the finding, and through the rebuttal period, the authors provided a number of experiments to compare against other variants of their baselines (and ablations), with some reviewers increasing their scores in favor of acceptance.

At the same time, there was concern that the U-Net architecture could potentially reconstruct the movie without any retinal data. Thus, it was not entirely clear whether the model was truly leveraging the retinal data to obtain meaningful outputs. Unfortunately, this concern was not fully addressed in the revision, leaving the reviewers overall with mixed views but the majority in favor of acceptance.


**Award:**

No

---

### Decision · Program_Chairs · 2022-09-14

Accept